# Biopsychosocial Correlates of Quality of Life in Multiple Sclerosis Patients

**DOI:** 10.3390/ijerph192114431

**Published:** 2022-11-04

**Authors:** Ana Rita Batista, Susana Silva, Leonor Lencastre, Marina Prista Guerra

**Affiliations:** Center for Psychology at University of Porto (CPUP), Faculty of Psychology and Education Sciences, University of Porto, 4200-135 Porto, Portugal

**Keywords:** multiple sclerosis, quality of life, biopsychosocial, meaning in life

## Abstract

Multiple sclerosis (MS) is a demyelinating chronic disease that has had increasing prevalence over the last years. We have investigated whether the perceived quality of life is reduced in multiple sclerosis patients compared to control participants with a cross-sectional approach, and how it relates to sociodemographic, clinical, and psychosocial variables in MS with multiple regression. To that end, a group of MS patients (*n* = 50) and a control group (*n* = 50) that was matched for age and education level filled in the WHOQOL-BREF (perceived quality of life across four domains) and a sociodemographic questionnaire. The participants in the MS group also filled in a clinical questionnaire and three instruments measuring psychosocial variables (the DASS-21 for depression, anxiety, and stress, the Brief-COPE for coping skills, and the Meaning in Life Scale). The results showed that the perceived quality of life was lower in the MS group than in the control group. Multiple regression models incorporating the variables that showed significant correlations with the quality of life indicated that age, professional status, recovery from relapses, depression, active coping, and meaning in life predicted at least one domain of the quality of life. Meaning in life predicted the quality of life in all four of the domains. Although the quality of life in MS is linked to multiple biopsychosocial variables, meaning in life seems crucial.

## 1. Introduction

Multiple Sclerosis (MS) is a chronic autoimmune disease of the central nervous system that is characterized by inflammation, demyelination, and neurodegeneration [1]. It is the most common cause of non-traumatic disability in young adults [1,2,3]. Epidemiological data from 2020 have referred to 2.8 million MS patients worldwide [3]. Although the first symptoms of MS may appear at any age, most patients are diagnosed between 20 and 50 years old [3,4]. The prevalence of MS increases with the geographical distance from the equator (though exceptions exist) and it is two to three times higher in women [2,3].

The emergence, the symptoms, and the clinical evolution of MS are heterogenous [1]. The etiology of the disease is not yet fully understood, but the interaction between environmental and genetic factors seems to be key [2]. MS-related symptoms and disabilities depend on the location and the extent of demyelination. Therefore, they differ not only across patients, but also across the different stages of the disease in a single individual [5].

The impact of MS on life expectancy is not significant; however, since there is currently no cure for MS, patients tend to live with the disease for several decades [3,6,7]. Consequently, MS patients often face great challenges regarding their life prospects, their employment perspectives, and their expectations for the future. Even when patients succeed in overcoming previous relapses (or exacerbations or attacks), new relapses bring in new limitations and, thus, a further need to adapt to the novel circumstances [7]. Thus, the permanent challenges that MS patients must face have a significant negative effect on their quality of life [6,7,8].

Quality of life (QoL) is a multidimensional construct that comprises the following four domains: physical, psychological, social, and environmental [9]. QoL is related to multiple variables, including sociodemographic, clinical, and psychosocial correlates [4]. Determining the correlates of QoL in MS patients may contribute to the better planning of interventions, treatments, and services [10]. Several studies have addressed the correlates of the QoL in MS; however, these are scattered across studies and some variables show mixed results.

Among the sociodemographic variables, age has consistently been shown to have no significant relationship with the QoL in MS patients [4,8,11]. Concerning professional status, Papuć and Stelmasiak [8] found that patients with an active professional status show improved QoL compared to non-active patients. Finally, the available findings regarding education level are mixed; while Strober [4] and Papuć and Stelmasiak [8] found no evidence of association with QoL, Zengin et al. [12] reported increased QoL in all four of the domains in patients with higher education levels.

Among the clinical variables, the research has focused on the current treatment, the disease type, the number of previous relapses, the presence of cognitive impairment, and the disease duration. The results concerning the current treatment type were null. Stuchiner et al. [13] found that patients who moved from injectable treatment to oral treatment showed no evidence of changes in QoL when they were compared with patients who remained on injectable treatment. Regarding the disease type, there are four types of MS, as follows: relapsing-remitting (RRMS), secondary-progressive (SPMS), primary-progressive (PPMS), and benign [3,7,14]. Papuć and Stelmasiak [8] found that the QoL is higher in RRMS than in SPMS and PPMS, which may be due to the faster progression of the disease in SPMS and PPMS. A significant relationship with number of previous relapses has also been reported; a higher frequency of relapses since diagnosis is associated with a lower QoL [15]. Finally, Benito-León et al. [16] found that the presence of neurocognitive impairment correlates negatively with all of the QoL domains. Mixed results have been found for the disease duration; while some studies found no significant association of this variable with QoL [8,11,17], Szilasiova et al. [18] showed decreases in QoL as the disease progresses.

The negative association of the QoL with psychosocial variables such as anxiety, depression, and stress is well established. The chronic nature of MS, its uncertain prognosis, and the absence of a definite therapy (i.e., a cure) contribute to these psychological symptoms [11,19,20]. MS patients show increased levels of anxiety and depression compared to healthy participants, and these conditions affect more than 20% of patients [8,21,22,23]. Both anxiety and depression correlate negatively with QoL, and they appear to be significant predictors in regression models. Depression has been observed to be the strongest predictor in several studies [4,8,10,11,17,20,24,25,26]. Regarding stress, Karimi et al. [27] found a high prevalence of stress in MS patients, with 23% presenting severe stress and almost 45% presenting moderate stress. The relationship between the QoL and stress has been investigated less than that with anxiety and depression. Nevertheless, both Wollin et al. [28] and Salehpoor et al. [20] reported a negative correlation between stress and QoL.

The literature on the psychosocial variables also highlights the association between the QoL and coping strategies. Holland et al. [29] found that most MS patients use a combination of problem- and emotion-focused coping strategies, namely acceptance, planning, positive reinterpretation and growth, and active coping. Strober [4] reported that patients with a lower QoL adopt maladaptive coping strategies, such as behavioral disengagement and denial, while those with a higher QoL tend to use adaptive and problem-focused coping strategies, such as planning, active coping, emotional and instrumental social support, humor, and positive reinterpretation and growth. Similar results were obtained by Zengin et al. [12], as follows: a positive correlation was observed between QoL and problem-focused strategies, specifically active coping, planning, positive reframing, and acceptance. Emotion-focused strategies, such as denial, substance use, and self-distraction, showed negative correlations with QoL.

Finally, meaning in life is a relevant psychosocial correlate of physical and mental health, and it is associated with an increased QoL in chronic diseases, such as cancer [30,31,32] and HIV/AIDS infection [33]. The literature on meaning in life and MS is not abundant, but there is evidence that various QoL domains show increased values with increased meaning in life, suggesting that this variable may foster adaptation to the disease [24,26].

The correlates of the QoL do not necessarily constitute causes of it, and—except for age—we cannot rule out that the QoL has a causal impact on the variables that we have mentioned. However, if causal direction goes from the correlates of the QoL, targeting the relevant biopsychosocial factors of the QoL is fundamental to the successful adaptation of patients. As we observed in the literature review, most of the studies focused on a narrow set of correlates of the QoL in MS, and some of the correlates have displayed mixed results. Therefore, our goals were to determine whether the quality of life (QoL) is reduced in MS patients compared to control participants, and to specify how the quality of life relates to the sociodemographic, clinical, and psychosocial variables in MS.

## 2. Materials and Methods

### 2.1. Participants

A group of MS patients (clinical group, *n* = 50) and a control group with healthy participants (*n* = 50) took part in this study. Sensitivity power analyses showed that this sample size was able to capture the medium (*d* = 0.50) effect sizes with 80% power and a critical alpha of 0.05 when comparing the two groups. Data collection was carried out through an online questionnaire. The clinical group (examined for QoL and sociodemographic variables) included members of two multiple sclerosis associations, while the control participants (examined for QoL, sociodemographic, clinical, and psychosocial variables) were recruited in order to match with the MS participants for gender, age, and level of education.

As shown in Table 1, the two groups did not differ significantly in any variable, except professional status (more MS patients were non-active compared to the controls). Nevertheless, effect sizes—including that for professional status—were small (Jacob and Cohen, 1988).

Most participants of the MS group indicated a diagnosis of RRMS (54%), 10% presented SPMS, 10% had benign MS, 8% had PPMS, and 18% of the participants did not know their MS type. The mean disease duration ranged from 2 months to 30 years, with a mean of 8 years (SD = 7.54). Approximately half of the participants (42%) were on the injectable modality and the other half (48%) were taking oral medication (10% of the participants were not receiving treatment). Concerning the number of relapses that the participants had had since their diagnosis, the most frequent response category was ‘some’ (2–4 relapses, 40% of participants), while 28% had had many relapses (more than four), and 26% participants had had few (none or one) relapses since their diagnosis. Concerning recovery from relapses, the dominant response (56%) was ‘partially recovered’, and the other 44% participants recovered completely. Finally, 44% of participants reported cognitive impairment (as opposed to not having cognitive impairment, 34%, and not knowing whether they had, 22%).

### 2.2. Measures

All of the participants filled in a sociodemographic questionnaire for data on age, birth date, gender, marital status, level of education, and professional status. The clinical questionnaire, which was administered only to the clinical group, addressed MS patients’ medical history. It contained questions on disease type, duration of illness, current treatment, number of previous relapses and if they had partially or completely recovered, and the presence of neurocognitive impairment.

To evaluate QoL, we used World Health Organization - Quality of Life - BREF (WHOQOL-BREF), which was designed by the WHOQOL Group [34]. The instrument was adapted for the Portuguese population by Vaz Serra et al. [35]. The WHOQOL-BREF is a self-report tool that comprises 26 items. Two of these address general QoL, and the other 24 cover the following four specific domains: physical, psychological, social, and environmental. The score obtained in each domain ranges between 0 and 100, with higher scores representing higher perceived QoL. The WHOQOL-BREF has a good internal consistency [35]. In the present study, Cronbach’s alpha was 0.94 for all items, 0.87 for the physical domain, 0.86 for the psychological domain, 0.84 for the social domain, and 0.84 for the environmental domain. These values indicate very good internal consistency in our sample.

The levels of anxiety, depression, and stress were assessed with the Depression Anxiety Stress Scale—21 items (DASS-21), which is a self-report instrument that was developed by Lovibond and Lovibond [36] and adapted for the Portuguese population by Pais-Ribeiro et al. [37]. It includes three scales, each composed of seven items (total of 21 items). Each item presents a statement that the participants must rate using a scale from 0 to 3. Higher scores indicate more negative affective states. For each scale, the scores may range from 0 to 21 [37]. In the present study, the internal consistency for this instrument was very good: Cronbach’s alpha was 0.88 for the anxiety scale, 0.91 for depression, and 0.90 for stress.

Coping strategies were accessed by Brief-COPE, which is an instrument that was created by Carver [38] and adapted for the Portuguese population by Pais-Ribeiro and Rodrigues [39]. It evaluates the frequency with which participants use a variety of coping strategies to deal with stressful or problematic situations. The instrument contains 14 scales, each with two items. The items measure coping reactions, some of which are considered adaptative and others maladaptive. The inventory has a total of 28 items. For each, the participants respond on a scale of 0 to 3 [39,40]. In the present study, the internal consistency values were the following: active coping, α = 0.66; planning, α = 0.82; using instrumental support, α = 0.78; using emotional support, α = 0.84; using religion, α = 0.88; positive reframing, α = 0.87; venting, α = 0.81; denial, α = 0.77; behavioral disengagement, α = 0.67; substance use, α = 0.68; and using humor, α = 0.92. The self-blame, acceptance, and self-distraction scales presented alpha values below 0.60, and, thus, they were not considered in the present study. The final results are presented per scale, and a global score is not available. Therefore, the scores may range between 0 and 6. Higher scores represent the coping strategies that the participant uses more often [39].

To measure the meaning in life, we used the Meaning in Life Scale (ML), which was developed by Guerra et al. [31]. The instrument comprises seven items, with a five-point scale for response. The global score is obtained by summing the scores for all items and considering that there are reverse items. The global score may range between 7 and 35. Higher scores indicate increased self-perceived meaning in life. This scale presents adequate values of internal consistency [31]. In the present study, the internal consistency was good (α = 0.76).

### 2.3. Procedure

The project was approved by the local ethics committee (Ref.202/09-5b; date of approval: 14 December 2020). Data collection was carried out with an online questionnaire. The participants were given a link to access the form. The first part presented information about the study, namely information on goals, anonymity, and confidentiality. Before any response, the participants were asked to provide informed consent. The questionnaire compiled all of the instruments into a single form. The questionnaire took approximately 15 min to be completed in the MS group, and 5 min in the control group.

Data collection for the MS group took place between December 2020 and February 2021. Members of Associação Nacional de Esclerose Múltipla (National Multiple Sclerosis Association) and Associação Todos com Esclerose Múltipla (All With Multiple Sclerosis Association), which are two Portuguese associations for MS, volunteered to participate. The inclusion criteria were being older than 18 and having been diagnosed with MS. Data collection for the control group took place later, in February 2021. The control participants were given the same information as the MS group, and they also provided informed consent. In this form, only the sociodemographic questionnaire and the WHOQOL-BREF were included. Members of the population that were known to the team were recruited for the control group, such that they could be matched with MS participants for gender, age, and level of education (snowball method [41]).

### 2.4. Data Analysis

Data were analyzed using *Statistical Package for the Social Sciences* (IBM SPSS Statistics 27). Parametric tests were used whenever the assumption of normality was verified, and non-parametric tests were used otherwise. The critical alpha levels were set to 0.05, with confidence intervals of 95%.

The correlates of QoL were first investigated one by one, using either correlations or tests for group comparisons (t-Student for independent samples, one-way ANOVA, Kruskal–Wallis, Mann–Whitney U) depending on the variable type (continuous vs. categorical). Effect sizes were classified following Cohen [42], where values below 0.30 define small effects, values between 0.30 and 0.50 define medium effects, and those above 0.50 as large. We then defined three-predictor regression models (the maximum number of predictors allowed with the current sample size, according to the 15-participants-per-predictor rule [43]) for each QoL domain. The predictors were selected according to the following criteria: first, they should be significantly associated with the dependent variable (for continuous predictors, the correlations should be above 0.30 and below 0.90, and the strongest associations should be preferred), as stated by Tabachnick and Fidell [44]; second, at least one predictor should be clinical or sociodemographic. Using these models, we were able to validate the associations that we found in the first phase, better determine the relative weight of different correlates, and also quantify how much each set of correlates explained the variability in QoL.

## 3. Results

### 3.1. QoL in MS vs. Controls

The control group (*n* = 50) showed significantly higher levels of QoL for general QoL, as well as for the physical, the psychological, and the social domains, than the clinical group (*n* = 50; Table 2). The effect sizes were large. For the environmental domain, cross-group differences did not reach significance, and the effect size was moderate. After Bonferroni corrections for multiple comparisons, the pattern of results was preserved (*p*s < 0.005).

### 3.2. Descriptive Results of Psychosocial Variables

Anxiety (*M* = 5.24, *SD* = 4.73), depression (*M* = 5.46, *SD* = 4.93), and stress (*M* = 7.62, *SD* = 4.68) values of the clinical group (*n* = 50) ranged between 0 and 18 and 20 and 21, respectively. Active coping (*M* = 3.58, *SD* = 1.49), planning (*M* = 3.54, *SD* = 1.59), using instrumental support (*M* = 2.40, *SD* = 1.71), using emotional support (*M* = 2.90, *SD* = 1.74), religion (*M* = 2.18, *SD* = 2.09), positive reframing (*M* = 3.56, *SD* = 1.53), venting (*M* = 2.76, *SD* = 1.77), denial (*M* = 1.52, *SD* = 1.59), behavioral disengagement (*M* = 1.26, *SD* = 1.41), and humor (*M* = 2.82, *SD* = 1.85) ranged between 0 and 6, while substance use (*M* = 0.36, *SD* = 0.94) ranged between 0 and 4. The minimum in the Meaning in Life Scale (*M* = 25.50, *SD* = 4.59) was 18 and the maximum was 35.

### 3.3. Correlations between Psychosocial Variables and QoL in MS

Table 3 shows the correlations between the QoL and the psychosocial variables for the MS group (*n* = 50). Meaning in life, active coping, and humor showed positive associations with QoL, while anxiety, depression, stress, behavioral disengagement, and substance use showed negative associations.

### 3.4. Associations of Sociodemographic and Clinical Variables with QoL in MS

Within the sociodemographic variables, age and professional status, but not the level of education, showed significant associations with the QoL in the clinical group (*n* = 50). Age correlated negatively and moderately with the social domain of QoL, *r*(50) = −0.31, *p* = 0.031, but not with the other domains. Having an active professional status was linked to significantly increased QoL levels, with large effect sizes in all of the domains (the general domain: *t*(48) = 2.79, *p* = 0.008, *d* = 0.79; the physical domain: *t*(48) = 6.62, *p* < 0.001, *d* = 1.87; the psychological domain: *t*(48) = 3.90, *p* < 0.001, *d* = 1.11; and the environmental domain: *t*(48) = 3.12, *p* = 0.003, *d* = 0.88), except the social domain (*p* = 0.075, *d* = 0.51).

Within the clinical variables, the disease type, the number of relapses, the recovery from relapses, and the neurocognitive alterations showed significant associations with the QoL. Concerning the disease type, the Kruskal–Wallis tests showed significant (though with a small effect size) differences in the environmental domain of the QoL, depending on the MS types, *X*^2^(4, 50) = 11.1, *p* = 0.025, *η*^2^ = 0.23. The Mann–Whitney U tests were used to clarify the differences between SPMS, which was the type with the lowest QoL values (*Mdn* = 43.8, *n* = 5), and the other three MS types, with the following results: RRMS (*Mdn* = 65.6, *n* = 27), *U* = 10.5, *z* = −2.97, *p* = 0.003, *r* = −0.42, PPMS (*Mdn* = 62.5, *n* = 4), *U* = 0.000, *z* = −2.47, *p* = 0.014, *r* = −0.34, and benign MS (*Mdn* = 65.6, *n* = 5), *U* = 3.00, *z* = −2.02, *p* = 0.044, *r* = −0.28. The effect sizes were moderate. The remaining domains of QoL did not present any differences.

The participants with a higher number of relapses showed lower levels of QoL in the physical domain, with a small effect size, *F*(2,44) = 4.03, *p* = 0.025, *η*^2^ = 0.16. Using post-hoc (Tukey) tests, we found significant differences between few relapses (*M* = 74.5, *SD* = 22.3) and many relapses (*M* = 53.8, *SD* = 20.1). In the other QoL domains, the differences were non-significant and showed small effect sizes (*p*s > 0.067, *η*^2^s < 0.37). Regarding the recovery from relapses, significant differences between complete and partial recovery emerged for general, physical, social, and environmental domains, with large effect sizes (the general domain: *t*(48) = 2.96, *p* = 0.005, *d* = 0.84; the physical domain: *t*(48) = 3.00, *p* = 0.004, *d* = 0.85; the social domain: *t*(48) = 2.50, *p* = 0.016, *d* = 0.71; and the environmental domain: *t*(48) = 2.70, *p* = 0.010, *d* = 0.77). The participants with partial recoveries presented lower QoL levels. For the psychological domain, the differences were non-significant (*p* = 0.055, *d* = 0.56).

Finally, the participants who had cognitive impairment (*M* = 52.6, *SD* = 16.3) differed significantly from those who did not have any impairment (*M* = 66.4, *SD* = 21.9) in the physical domain of the QoL, *t*(37) = −2.25, *p* = 0.030, *d* = −0.72, with a large effect size. The remaining domains did not present any differences, and the effect sizes were small (*p*s > 0.20, *d*s < −0.42).

The disease duration and the type of treatment were unrelated to the QoL. Regarding the disease duration, no correlation was found (*p*s > 0.076). In respect to the type of treatment (excluding the five subjects who were not under any treatment), no differences were found between the participants who were under oral treatment and those who were receiving injectable treatment, in all of the QoL domains (*p*s > 0.20, *d*s < 0.39).

### 3.5. Regression Models with Selected Predictors

The three-predictor multiple regression models for each QoL domain of the clinical group (*n* = 50), resulting from the criteria that we used (the strongest significant associations with the QoL, at least one clinical or sociodemographic predictor) are presented in Table 4.

All five of the models showed substantial coefficients of determination (*R^2^* > 0.26). In the model that was used for the physical domain, the professional status was the strongest predictor. Meaning in life was the strongest predictor in the psychological, environmental, and social domains. In the model for the general QoL, meaning in life also played an important role; although, it was not superior to that of recovery from relapses and active coping.

## 4. Discussion

Our goals were to determine whether quality of life (QoL) is reduced in MS patients compared to control participants, and to specify how the quality of life relates to the sociodemographic, clinical, and psychosocial variables in MS.

Regarding the first goal, we saw lower levels of QoL in the MS patients compared to the controls for all of the domains, except for the environmental domain. These findings are in line with the literature [6,8], and they likely reflect these patients’ need to adjust to new life scenarios and the difficulties they have experienced while living with the disease [7]. Since MS patients tend to have poorer physical security, financial resources, home environments, and access to transportation (items that are covered by the environmental domain of WHOQOL-BREF), significantly lower levels of environmental QoL could also be expected in the MS patients compared to the controls. This was not the case, even though the MS patients’ values were lower than those of the controls. One explanation may be that our MS participants were all members of associations, where support is provided in these areas. Future research could, thus, consider membership in associations as a potential moderating factor of environmental QoL. As a side note on the QoL values of the MS patients, we found moderate values in all of the domains, which is in line with the findings of Pinto and Guerra [26]. The general domain showed the lowest score, while the environmental domain showed the highest score.

When pursing our second goal—determining how QoL relates to biopsychosocial factors in MS patients—we characterized the psychosocial variables in these patients. Regarding anxiety, depression, and stress, our sample showed moderate negative values, which is not totally consistent with the literature [8,21,22,23,27]. Once again, a likely explanation could be our participants’ membership in associations that provide psychological services and, hence, attenuate the psychosocial impact of the disease. Concerning the coping strategies, our patients reported a combination of problem- and emotion-focused strategies, which is in line with what has been highlighted in the literature [12]. The most used strategies of our participants were active coping, planning, and positive reframing, which parallels the findings of Holland et al. [29]. These strategies are considered to be adaptative, in the sense that they are linked to better adaptation to the disease [29]. For meaning in life, our MS patients reported positive values, which were similar to those that were obtained by Pinto and Guerra [26] in a similar study (*M* = 25.37, *SD* = 4.97). However, when they were compared with the healthy participants from other studies [31] (*M* = 28.10, *SD =* 3.71), the MS patients’ values are lower.

The analysis of the psychosocial correlates of the QoL highlighted the importance of meaning in life; however, the other variables were also relevant. In line with the literature, anxiety and depression were associated with lower QoL in several domains [8,10,11,20,25,26]. Stress correlated negatively with the psychological domain of QoL, strengthening the previous findings [20,28]. The fact that anxiety, depression, and stress did not correlate with all of the QoL domains may relate to the low variability that we had for these measures. In the regression model for the psychological domain, depression was not the strongest predictor of QoL, which goes against some of the available findings [4,8,10,11,17,20,24,25,26]. Nevertheless, it was a significant predictor, indicating that a timely diagnosis and an adequate intervention in depression may be crucial in order to improve the QoL in MS patients.

Concerning the coping strategies as the psychological correlates of QoL, our findings were in line with the literature [4,12] by pointing to a positive association of QoL with active coping and humor strategies, and a negative correlation with behavioral disengagement and substance use. According to Meyer’s [45] classification of the coping strategies as adaptive vs. maladaptive, the pattern that we found points to the association between the adaptive strategies and higher QoL, and vice-versa, which is in line with Strober [4]. Another way to summarize our findings is to consider the duality approach coping (to try to solve the problem, such as active coping) vs. avoidance coping (to avoid the problem and disengage, such as behavioral disengagement), which was proposed by Nes and Segerstrom [46], and to link approach coping to higher levels of QoL. In contrast, the opposition between emotion- and problem-focused coping is not heuristic in our case, since both substance abuse (which is linked to lower QoL) and humor (which is linked to higher QoL) are both considered to be emotion-focused strategies. Active coping was a significant predictor in the regression model for general QoL—it was even the strongest predictor in this model. Active coping is an adaptative, problem-focused, approach strategy, and it overlaps with the idea of mobilizing resources in order to deal a problem [45,46]. In this view, it makes sense that active coping has a privileged association with general QoL. Promoting the use of this strategy by MS patients may, therefore, contribute to improving their QoL.

As has been mentioned above, meaning in life had a dominant role among the psychosocial correlates of QoL. Higher scores of meaning in life were associated with higher levels of general, physical, psychological, social, and environmental QoL. Moreover, meaning in life was a significant predictor in all of the regression models, and it was the strongest predictor in the psychological, social, and environmental domains. Though the literature on meaning in life and QoL is scarce, our findings match those of Pinto and Guerra [26] and Oliveira [24]. The relevance of meaning in life for the QoL of MS patients parallels what happens in other pathologies [31,33,47].

The analysis of the sociodemographic correlates of QoL pointed to the relevance of age and professional status, but not the level of education. Age correlated negatively with social QoL. This finding does not support those from previous studies on MS, where age did not correlate with QoL [4,8,11]. Nevertheless, this parallels what happens in the healthy population, where social isolation increases with age and, thus, potentially impacts social QoL. Concerning professional status, we found that an active status relates to higher QoL in the physical, psychological, and environmental domains, with the strongest association occurring with the physical domain. These findings corroborate those from Papuć and Stelmasiak [8], and they make sense in light of the fact that the patients with lower levels physical QoL are those who cannot keep their capacity to have an active life; many MS patients stop working prematurely, reduce their working hours, or change their jobs as a result of physical fatigue [48]. However, since an active professional status is crucial to one’s identity as part of a community, it is important to promote an active status in MS patients for as long as possible. To that end, awareness must be raised in employers [49].

Finally, we have found the following four clinical correlates of the QoL in MS: the disease type, the number of previous relapses, the recovery from relapses, and neurocognitive impairment. Regarding the disease type, the patients with SPMS showed lower QoL than the other groups. This is consistent with the literature, which points to a lower QoL in progressive types, such as SPMS or PPMS, compared to RRMS [8]. The correlation was, however, restricted to the environmental domain. An increased number of previous relapses was linked to lower physical QoL. This is an expected result, in that relapses, by definition, degrade the physical condition of patients, as shown by Mäurer et al. [15]. Regarding the recovery from relapses, the participants with complete recovery showed increased QoL in the general, physical, social, and environmental domains. Recovery from relapses was also a significant predictor of the general domain QoL. Naturally, an incomplete (partial) recovery has a negative impact on all areas of life [15], and, thus, these findings make sense. Finally, regarding cognitive impairment, we found a negative correlation with the physical QoL only, unlike Benito-León et al. [16], who found negative correlations for all of the domains. The relationship with the physical domain is understandable in light of the topics that are covered by the physical domain items (the capacity for daily life activities and working capacity). As for the reason why neurocognitive impairment did not relate to the other QoL domains, it may be a methodological one; unlike Benito-León et al. [16], not all of our patients had a formal diagnosis, and the responses they provided on neurocognitive impairment reflected their own perception—which might not correspond to reality. As for the null relationship of QoL with the disease duration, it is in line with previous results [8,11,17]. The lack of correlations with the treatment type is also consistent with the literature [13].

Although our study has made important contributions to clinical practice and to future research, it is not exempt from limitations. One of these concerns the sample that we used—a convenience sample, consisting only of women. Although women represent the majority of MS patients, men also suffer from the disease. Therefore, it is important that future research promotes more random sampling methods, as well as including both men and women, in order to analyze gender effects. In addition—and although our sample was large enough to capture the effect sizes in the upper limit of the medium range—it could be important to increase the sample size in future research. Another limitation arises from the online data collection method, which did not allow us to get a medical confirmation of neurocognitive impairment and left us with the participants’ own perceptions on this matter. Finally, the control and the MS group differed in the distribution of professional status, and we found that the professional status affects the QoL. This raises a question on the extent to which the cross-group differences that we found for the QoL are, at least partly, due to this difference. Concerning this, it should be noted that the effect sizes for this difference were small; however, it could be worth improving the group matching for this variable in future studies.

Our study has contributed to a better understanding of the QoL in the MS population—a group where QoL has not received much attention thus far. Most of our findings were in agreement with the available literature, but a few novel findings have emerged, raising new questions for future research. For the correlates of QoL that have presented mixed results (null vs. negative relation with QoL)—the education level and the disease duration—our study supported the absence of relationships. For age as a potential correlate of QoL, our results have suggested that—in contrast to previous research—it may associate negatively with the QoL in the social domain. The psychosocial characterization of our MS sample suggested the presence of low levels of anxiety, stress, depression, maladaptive coping strategies, or problems in finding meaning in life, which might be due to the fact that all of our participants received psychological support from associations. Meaning in life showed extremely clear associations with the QoL—stronger and wider associations than any other correlate that has been considered here. All of these findings deserve future research attention.

## 5. Conclusions

In sum, our study has contributed to strengthen the idea that the QoL is diminished in MS patients compared to controls. Among all of the correlates, meaning in life had a particularly strong presence, suggesting the need for interventions that target this goal. The relevance of the clinical and sociodemographic correlates, along with the psychosocial correlates, stress the need for holistic and multidimensional interventions in MS.

## Figures and Tables

**Table 1 ijerph-19-14431-t001:** Comparison of MS and healthy participants on sociodemographic variables (N = 100).

Variables	Clinical Group (N = 50)	Control Group (N = 50)	Group Comparisons				
			*X* ^2^	*t*	*df*	*p*	Effect Size
Age							
Mean ± SD * (years) Minimum–Maximum	38.58 ± 9.54 18–68	39.16 ± 8.11 19–60		0.328	98	0.744	0.066 ^1^
Education level							
Mean ± SD * (years) Minimum–Maximum	14.52 ± 2.83 9–19	14.98 ± 3.52 4–23		0.720	98	0.473	0.144 ^1^
Gender, N (%)							
Male Female	0 (0%) 50 (100%)	0 (0%) 50 (100%)					
Marital status, N (%)							
Single Married/partnered Divorced/separated Widowed	15 (30%) 27 (54%) 8 (16%) 0 (0%)	10 (20%) 35 (70%) 5 (10%) 0 (0%)	2.725		2	0.256	0.165 ^2^
Professional status, N (%)							
Active Non-active	26 (52%) 24 (48%)	37 (74%) 13 (26%)	4.290		1	0.038	0.228 ^3^

* SD: standard deviation. ^1^ Cohen d. ^2^ Cramér V. ^3^ Pearson’s Phi.

**Table 2 ijerph-19-14431-t002:** Descriptive statistics for QoL domains (means, standard deviations, and range) and comparison of MS participants and healthy participants on QoL scores.

Variables	Clinical Group (N = 50)		Control Group (N = 50)		Group Comparisons			
	**Mean ± SD ***	**Minimum–** **Maximum**	**Mean ± SD ***	**Minimum–** **Maximum**	* **t** *	* **df** *	* **p** *	* **d** *
General	58.75 ± 22.76	12.50–100.00	71.75 ± 15.11	25.00–100.00	3.364	85.167	0.001	0.673
Physical	61.50 ± 20.55	17.86–100.00	77.79 ± 11.93	50.00–100.00	4.846	78.648	<0.001	0.969
Psychological	60.33 ± 17.68	20.83–95.83	73.00 ± 16.95	25.00–95.83	3.657	98	<0.001	0.731
Social	59.83 ± 23.00	0.00–100.00	78.67 ± 14.98	50.00–100.00	4.851	84.221	<0.001	0.970
Environmental	66.19 ± 15.79	40.63–100.00	71.31 ± 13.10	43.75–96.88	1.766	98	0.081	0.353

* SD: standard deviation.

**Table 3 ijerph-19-14431-t003:** Descriptive statistics for QoL domains (means, standard deviations, and range) and comparison of MS participants and healthy participants on QoL scores.

	General Domain	Physical Domain	Psychological Domain	Social Domain	Environmental Domain
Anxiety	−0.150	−0.406 **	−0.317 *	−0.200	−0.176
Depression	−0.173	−0.329 *	−0.501 **	−0.297 *	−0.232
Stress	−0.129	−0.245	−0.442 **	−0.235	−0.227
Meaning in Life	0.463 **	0.503 **	0.661 **	0.513 **	0.608 **
Active Coping	0.390 **	0.429 **	0.321 *	0.238	0.350 *
Planning	0.191	0.227	0.127	0.042	0.135
Using Instrumental Support	0.157	0.249	0.066	−0.024	0.225
Using Emotional Support	0.068	0.012	−0.162	−0.153	−0.047
Religion	0.229	−0.100	−0.132	−0.094	−0.052
Positive Reframing	0.238	0.227	0.153	0.024	0.214
Venting	0.250	0.230	0.043	0.009	0.124
Denial	0.076	0.165	−0.050	−0.138	−0.151
Behavioral Disengagement	0.007	−0.052	−0.301 *	−0.237	0.042
Substance Use	−0.281 *	−0.267	−0.396 **	−0.284 *	−0.374 **
Humor	0.244	0.363 **	0.331 *	0.138	0.309 *

* *p* < 0.05. ** *p* < 0.01.

**Table 4 ijerph-19-14431-t004:** Multiple regression models for the QoL domains.

Regression Model	*β*	*t*	*p*	Adjusted *R*^2^	*F*(3,46)	*p*	Part *r*^2^
General Domain				0.331	9.069	<0.001	
Recovery from relapses	−0.299	−2.450	0.018				0.082
Active coping	−0.299	2.464	0.018				0.083
Meaning in life	0.297	2.342	0.024				0.075
Physical Domain				0.587	24.239	<0.001	
Professional status	−0.554	−5.636	<0.001				0.267
Active coping	0.182	1.856	0.070				0.029
Meaning in life	0.302	3.096	0.003				0.081
Psychological Domain				0.582	23.742	<0.001	
Professional status	−0.285	−2.912	0.006				0.072
Depression	−0.276	−2.792	0.008				0.067
Meaning in life	0.499	5.024	<0.001				0.215
Social Domain				0.353	9.893	<0.001	
Age	−0.297	−2.581	0.013				0.088
Recovery from relapses	−0.212	−1.765	0.084				0.041
Meaning in life	0.448	3.738	<0.001				0.185
Environmental Domain				0.437	13.682	<0.001	
Professional status	−0.260	−2.337	0.024				0.063
Substance use	−0.202	−1.791	0.080				0.037
Meaning in life	0.475	4.081	<0.001				0.191

## Data Availability

The dataset used in this paper is available at https://osf.io/8f57h/?view_only=8e06c1977543462b9a35b5dd5159b54f.

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
