# Peer review of "Biopsychosocial Correlates of Quality of Life in Multiple Sclerosis Patients"

_ijerph, 2022, doi:10.3390/ijerph192114431_

Round 1

Reviewer 1 Report

In this study, the author reported that QoL is diminished in MS patients compared to controls, and meaning in life had a particularly strong presence, suggesting the need for interventions that target, and also to stress the need for holistic and multidimensional interventions in MS.

The followed questions need to be considered:

1. To make it more reasonable, the introducion part can be shorten.

2. Here, the number of clinical group is n = 50, and are all female, maybe a large number of patients will be better.

Author Response

Reviewer #1:

In this study, the author reported that QoL is diminished in MS patients compared to controls, and meaning in life had a particularly strong presence, suggesting the need for interventions that target, and also to stress the need for holistic and multidimensional interventions in MS.

The followed questions need to be considered:

  1. To make it more reasonable, the introduction part can be shorten.

R: We agree with the reviewer and the introduction part has been shortened.

  1. Here, the number of clinical group is n = 50, and are all female, maybe a large number of patients will be better.

R: We agree with the Reviewer, and this is why we highlighted the gender-related limitation of our sample in the discussion (lns 453-457). Concerning sample size, we ran sensitivity power analyses and found that our sample was large enough to capture effect sizes in the upper limit of the medium range (d = .50) for group comparisons. This indicates that we had reasonable power in our study. Nevertheless, we totally agree that this could be improved in future research. Therefore, we have now added this topic to the discussion (lns 457-459).

---

We are grateful to the Reviewers for their constructive comments, which helped us improving our manuscript.

Reviewer 2 Report

Overview

1. The paper is original. I would recommend publication after the following revisions by the authors.

Title and Abstract

2. I suggest adding the study design

Introduction

3. I suggest adding clear aims of the study at the end of the introduction.

Methods—Participants

4. lines 119-120 are descriptive results, not methods. I suggest moving those lines to the result section.

Methods—Procedure

5. This section could be moved to the beginning of the methods section.

6. if possible, mention the two associations that the patients were recruited from

7. more details about the recruitment of the controls are needed, such as where they were recruited from.  

Methods—Analysis  

8. In lines 226-229, on what bases are these criteria supported by the literature? Please give more details.  

Results

9. For each section and analysis, please specify your sample (entire sample or MS only) as well as the sample size.

10. Because of the multiple models/analyses, I suggest using a multiple-comparison correction method, such as Bonferroni correction or any other method.

Discussion

11. lines 321-322 are the aims of the study and could be moved to the end of the introduction. 

Author Response

Reviewer #2:

Overview

  1. The paper is original. I would recommend publication after the following revisions by the authors.

Title and Abstract

  1. I suggest adding the study design.

R: We followed your suggestion, and the abstract has been changed to integrate the study design.

Introduction

  1. I suggest adding clear aims of the study at the end of the introduction.

R: The aims of the study are now at the end of the introduction, and they have been clarified (lns 117-120).

Methods—Participants

  1. Lines 119-120 are descriptive results, not methods. I suggest moving those lines to the result section.

R: We understand that these lines (and the associated table, Table 1) may seem more appropriate to the results section. However, these lines were intended to justify the feasibility of the comparative analysis between the two groups, which is then presented in the results. Thus, in order to preserve the idea that these sociodemographic-comparative results have different roles within the study, we think it may be better to keep them in the Participants section.

Methods—Procedure

  1. This section could be moved to the beginning of the methods section.

R: We agree that the information provided in the Procedure section provides an important overview of the study and, therefore, reading can be facilitated if it is moved to the beginning. In order to address this without going against the traditional sequence of sections (participants, materials, procedure), we made a quick overview of the procedure at the beginning (lns 127-131) and kept the procedure part at the end of the method, with a more detailed description of data collection procedures.

  1. If possible, mention the two associations that the patients were recruited from.

R: The name of the two associations has been added in the procedure section (lns 230-232).

  1. More details about the recruitment of the controls are needed, such as where they were recruited from.  

R: Additional information about the recruitment of controls has been provided (lns 237-240).

Methods—Analysis  

  1. In lines 226-229, on what bases are these criteria supported by the literature? Please give more details.  

R: The number of predictors in multiple regression is based in Stevens [1], who suggested the ratio of 15 subjects for each predictor (in our case, for 50 participants, the number of predictors should be 3).

Also, according to Tabachnick and Fidell [2], regression models will be best when each independent variable is strongly correlated with the dependent variable but uncorrelated with other independent variables. This is the reason why we have selected the predictors we did, instead of others.

To clarify this point, we have added the references in those lines (lns 250-256).

  1. Stevens, J. Applied multivariate statistics for the social sciences, 4 ed.; Lawrence Erlbaum Associates: 2002.
  2. Tabachnick, B.G.; Fidell, L.S. Using multivariate statistics 4ed.; Allyn and Bacon: 2001.

Results

  1. For each section and analysis, please specify your sample (entire sample or MS only) as well as the sample size.

R: Additional information has been added in each section to specify the sample and sample size.

  1. Because of the multiple models/analyses, I suggest using a multiple-comparison correction method, such as Bonferroni correction or any other method.

R: We tested how Bonferroni corrections would affect group comparisons, and we found that the pattern of results was preserved (lns 265-266). Regarding the associations of clinical, sociodemographic and psychosocial variables with QoL, we did not apply these corrections because these results were merely subsidiary of our main, regression-based analysis (the goal behind investigating the associations was determining the most relevant predictors), and regression validated the relevance of such associations.

Discussion

  1. Lines 321-322 are the aims of the study and could be moved to the end of the introduction. 

R: We agree, and the aims have been moved to the end of the introduction (lns 117-120).

---

We are grateful to the Reviewers for their constructive comments, which helped us improving our manuscript.

Round 2

Reviewer 1 Report

I sugget to accept this manuscript.

Reviewer 2 Report

The paper has improved and could be published in the journal